# Evaluating Associations Between Drought and West Nile Virus Epidemics: A Systematic Review

**DOI:** 10.3390/microorganisms13122851

**Published:** 2025-12-15

**Authors:** Marie C. Russell, Desiree A. Bliss, Gracie A. Fischer, Michael A. Riehle, Kristen M. Rappazzo, Kacey C. Ernst, Elizabeth D. Hilborn, Stephanie DeFlorio-Barker, Leigh Combrink

**Affiliations:** 1Oak Ridge Institute for Science and Education (ORISE), Hosted by US Environmental Protection Agency, Research Triangle Park, NC 27711, USA; 2School of Natural Resources and the Environment, University of Arizona, Tucson, AZ 85721, USA; desireeandersen@arizona.edu (D.A.B.); gfischer@arizona.edu (G.A.F.); 3Lovejoy Center, Arizona Institute for Resilience, University of Arizona, Tucson, AZ 85721, USA; 4Department of Entomology, University of Arizona, Tucson, AZ 85721, USA; mriehle@ag.arizona.edu; 5Center for Public Health and Environmental Assessment, Office of Research and Development, US Environmental Protection Agency, Research Triangle Park, NC 27711, USA; 6Epidemiology and Biostatistics Department, Mel & Enid Zuckerman College of Public Health, University of Arizona, Tucson, AZ 85721, USA

**Keywords:** drought, West Nile virus, systematic review, environmental indicators, mosquito-borne disease

## Abstract

Human West Nile virus (WNV) infections can have severe neurological health effects, especially among those over 50 years of age. As changes in weather patterns lead to more frequent and intense droughts, there is a public health need for improved understanding of drought associated WNV risks. While multiple studies have reported an association between drought conditions and human WNV cases, this information has not yet been synthesized systematically across studies. Our review aims to evaluate the existing evidence of an association between drought and human WNV cases while considering the impacts of different study regions, methodological approaches, drought metrics, and WNV case definitions. We conducted a systematic literature search of peer-reviewed epidemiological studies that examined a potential association between drought and human WNV cases. Our inclusion criteria targeted studies that employed measures of drought beyond precipitation and reported effect estimates along with measures of error. The literature search and screening process resulted in the inclusion of nine papers with study periods spanning from 1999 to 2018. The included peer-reviewed publications employed a wide variety of study designs and methods, such as linear mixed-effects models, generalized linear models using simultaneous autoregression, generalized additive models, Bayesian model averaging, and a case-crossover design using conditional logistic regression models. We summarize the key findings and provide study quality evaluations for each of the nine included studies. Studies that analyzed drought indices averaged over a seasonal period of three to four months reported positive associations between drought and WNV. However, studies that analyzed drought indicator variables averaged over weekly periods of time had less consistent results. We discuss potential mechanisms underlying the observed associations between drought and human WNV cases.

## 1. Introduction

After West Nile virus (WNV) arrived in the United States (US) in 1999, it quickly emerged as the most common cause of epidemic meningoencephalitis in North America [1]. Approximately 70–80% of West Nile virus infections are asymptomatic. The disease manifests most commonly as a mild febrile illness accompanied by variable symptoms including headache, arthralgia, fatigue, gastrointestinal issues, and rash [2]; clinical presentations of these common WNV symptoms are likely to be misdiagnosed. The neuroinvasive form is estimated to occur in less than 1% of infected individuals [1], with higher risk among people over 50 years of age and an estimated 10% case fatality rate [3]. Hospital-based research suggests that more than one third of neuroinvasive WNV cases may not be properly diagnosed due to lack of testing or inappropriate testing [4].

With the potential for severe health impacts, there are concerns about how changing weather patterns may increase the risk of WNV transmission opportunities. Multiple studies have documented evidence of an association between drought and WNV epidemics [5,6,7,8] and increases in drought frequency and intensity have already been observed in certain regions of the world [9]. Within the US, the Great Plains and the Southwest are expected to have the greatest increase in climatic water deficit (CWD), which is the shortfall of water necessary to fully supply vegetation requirements [10]. The Great Plains region of the US has the highest incidences of human WNV neuroinvasive disease (WNND) by county based on case data from 1999 to 2023 [11]. However, in 2021, Maricopa County in Arizona experienced the largest focal WNV outbreak in a US county in recorded history, with 101 fatalities due to WNND [12]. At the height of the outbreak between July and September of that year, the monsoon season in Arizona had resulted in an over 200% increase in average precipitation in and around Maricopa County during what was characterized as long-term drought [13]. The decreased humidity that accompanies drought conditions is expected to reduce mosquito survival due to increased risk of desiccation, but mosquito biting rates are expected to increase under drier conditions [14]. The lifespan of female mosquitoes, mosquito density, and biting rate are known to impact vectorial capacity and thus influence disease transmission [15].

WNV is transmitted to humans primarily by mosquitoes of the *Culex* genus, including *Cx. pipiens*, *Cx. quinquefasciatus*, *Cx. restuans*, *Cx. molestus*, and *Cx. tarsalis* [16,17,18], with other genera such as *Aedes* and *Ochlerotatus* serving as potential bridge vectors between birds and humans [19,20]. The egg-laying strategy of *Culex* mosquitoes involves laying 100–300 eggs in a raft on the water’s surface, followed by hatching within 1–2 days [21]. The physiology and behavior of egg-laying female *Culex* mosquitoes may influence the relationship between water availability and WNV transmission to humans. For example, *Culex* mosquitoes actively avoid oviposition sites that have predator cues, including both predator kairomones and chemical alarm cues of mosquito prey [22], and drought conditions can intensify predation cues in shallow water at the edges of lakes and ponds [23]. When drought limits the overall number of suitable oviposition sites, *Culex* mosquitoes may engage in egg retention behavior [24] and relocate to more urban or suburban areas that have artificial containers as oviposition sites [23]. In addition, drought is likely to lead to increased desiccation stress in mosquitoes, and *Cx. pipiens* have been shown to increase their blood-feeding behavior to rehydrate when water availability is low [25]. These factors could increase the risk of WNV spillover to humans during a drought.

Another factor to consider is the circulation of WNV between mosquitoes and avian reservoir hosts. It has been suggested that when water resources are scarce, birds and mosquitoes gather in remaining areas with supportive greenspace and water sources, resulting in increased rates of contact and epizootic WNV transmission [6]. A recent study based in California, US found that out of 66 bird species, 27 species changed their habitat during drought conditions; species that shifted their habitat associations tended to leave natural habitats and use developed habitat and perennial agricultural habitat instead [26]. In rural areas with low human population density, the presence of irrigated agriculture can increase the risk of human WNV cases [27]. In suburban areas, residents can increase passerine abundance by gardening, maintaining vegetation, providing bird houses, filling bird baths, and stocking supplemental feeders with seeds [28]. In more urban settings, water resources may be connected to a combined sewer system, and *Culex* mosquitoes are attracted to water with high levels of organic material [29].

In this review, we summarize and evaluate previous analyses of the relationship between drought and human cases of WNV, paying particular attention to how drought is defined, the land use categories of each study region, and the way in which confounding variables and covariates are addressed. The objective of this study is to determine if there are any consistent patterns in the association between drought and human WNV cases across studies. We provide a comprehensive overview of different approaches published in the peer-reviewed literature to date that examined the association between drought and human WNV cases. A thorough investigation of this association is essential to predicting how the burden of vector-borne disease might increase in future years.

## 2. Materials and Methods

### 2.1. Identification of Relevant Search Terms and Databases

We identified search terms related to drought and cases of WNV and conducted our literature search on 17 October 2023, using PubMed and Web of Science databases (Appendix A). In previous studies that evaluated associations between drought exposures and human health outcomes, drought indices and definitions have been inconsistent [30]. Therefore, we kept our initial search broad by including multiple search terms related to drought, such as “arid”, “moisture”, “precipitation”, and “humidity” (Appendix A). The protocol for this review was registered with the International Prospective Register of Systematic Reviews (PROSPERO) prior to data extraction (CRD42024503289 [31]). Selection of included studies followed the PRISMA 2020 guidelines.

### 2.2. Screening and Data Extraction

We included peer-reviewed studies that reported an estimate of an association between a drought-related exposure and human cases of WNV. Each abstract identified from the literature search was screened by two reviewers, using the web-based systematic review software, SWIFT-Active Screener (SWIFT-AS) v1.21 [32]. Next, the full text of each article included after abstract-screening was screened by two reviewers through SWIFT-AS software. Review studies, meta-analyses, and predictive modelling articles that were relevant to our study objectives were marked during abstract-screening so that the references of these studies could be abstract-screened and full text screened if necessary (Figure 1). Because a previous study of climatic influences on WNV epidemics noted certain limitations of using precipitation as an explanatory variable [33], we only included studies that assessed a measure of moisture beyond precipitation as the exposure. In addition, studies that provided an effect estimate without reporting any measure of error around the estimate were excluded. Due to the rarity of neuroinvasive WNV cases, we did not exclude studies that reported only cases of WNV fever (WNF). In total, we identified nine studies that met our inclusion criteria (Figure 1). Data, such as title, year published, location, population, study period, study design, outcome, and effect estimate were extracted from each included article by one team member, and then the extracted data were reviewed for accuracy by two additional team members (Appendix A). The extracted data were evaluated to determine the feasibility of conducting a meta-analysis.

### 2.3. Study Quality Evaluation

The quality of each included study was evaluated using a modified version of the method implemented by the Office of Health Assessment and Translation [34], which is often used by US EPA’s Integrated Risk Information System [35]. Two reviewers rated each article based on criteria for seven different domains within US EPA’s Health Assessment Workspace Collaborative (HAWC) [36]. For each study, a score of “good”, “adequate”, “deficient”, or “critically deficient” was assigned to each domain: participant selection, exposure, outcome, confounding, analysis, selective reporting, and sensitivity. Score discrepancies were discussed and resolved between the two reviewers. An overall confidence rating of “high”, “medium”, “low”, or “uninformative” was also assigned to each study according to the guidelines presented in Appendix A.

## 3. Results

Of the nine studies that met our inclusion criteria, six were based in the US [8,37,38,39,40,41], and three were based in Europe and its neighboring countries [7,42,43]. Among the US-based studies, four used WNV case data from the CDC, but the types of data used still varied across these studies. Paull et al. [8] was the only study to analyze state-level WNND case data rather than county-level WNV data, including both neuroinvasive and non-neuroinvasive cases, and Soverow et al. [40] was the only study to make use of symptom onset dates at the case level. Two studies used case data from the European Centre for Disease Prevention and Control (ECDC) [7,43], but Tran et al. [43] supplemented ECDC data with case data identified through systematic review. The time periods examined by the nine included studies include years 1999 through 2018, and their main findings on a potential association between drought and WNV cases are presented in Table 1. Seven studies investigated whether drought conditions influenced human WNV cases during the same year [7,8,37,40,41,42,43], and two studies investigated whether interannual variations in drought conditions influenced human WNV cases during the second or third year [38,39] (Table 1). Among studies that investigated the association between drought and WNV cases during the same year, studies that used a multi-month drought duration period found evidence of a significant positive association between drought and human WNV cases [7,8], studies that used month-long averaging periods for drought indices also found some significant positive associations between summer drought and human WNV cases [37,41], but studies that used averaging periods of roughly one week had mixed results [40,42,43] (Figure 2). Of the nine included studies, one received a high overall confidence rating [40], four received medium overall confidence ratings [8,39,41,42], and four received low overall confidence ratings [7,37,38,43]; most low confidence studies were deficient in the confounding domain (Figure 3). Narrative summaries for the nine included studies can be found in Appendix A.

## 4. Discussion

A key insight emerging from this systematic review is that the temporal scale of drought measurement critically influences the observed relationship with human WNV cases, highlighting the importance of methodological considerations in climate-health research. To begin, methodologies of defining a positive WNV case must be standardized. Laboratory confirmation of WNV infection in humans is most often based on detection of WNV specific IgM and IgG antibodies in serum or cerebrospinal fluid; detection of the virus or RNA is difficult due to a short period of viremia [44]. While seven out of the nine included studies were rated as “good” for the exposure domain, only three out of nine were rated “good” for the outcome domain (Figure 3). Most of the studies included in this review combined WNF case data with WNND cases [37,38,39,40,41,42,43], but Paull et al. [8] only analyzed WNND case data, and Marcantonio et al. [7] only analyzed WNF case data. Paull et al. [8], Soverow et al. [40], and Stilianakis et al. [42] were rated “good” for the outcome domain because they either only used WNND cases, which are more likely to be accurately diagnosed than WNF, or they only used laboratory-confirmed human WNV cases. Among the studies that combined human WNV case data from multiple countries, Tran et al. [43] analyzed WNV data in units of outbreaks, rather than cases, to account for inconsistencies across data sources, and Marcantonio et al. [7] noted that “human WNF incidence data are limited by geographic variation in the accuracy of diagnosis, the establishment of surveillance, and the organization of national reporting systems”. Most studies used yearly totals of human WNV cases to reduce temporal autocorrelation [7,8,37,38,39,41,43], but two studies used weekly WNV incidence data and evaluated one to four week-long lag periods between the environmental exposure and the outcome [40,42].

The averaging periods for drought indices vary from months, to seasons, to years [45]. Among the studies that investigated whether drought conditions impact human WNV cases during the same year (Table 1, Figure 2), both studies that applied a seasonal averaging period of four months presented evidence of a significant association between drought and WNV cases [7,8]. Studies that applied an averaging period of one month produced results that tended to vary by season. While two studies reported a positive association between moisture and human WNV cases using March and April monthly averaged drought indices [37,41], the same two studies reported a significant association between drought and WNV cases when using data from later months, such as June, July, and August (Figure 2). These results are consistent with observations of higher-than-normal spring precipitation followed by dry summer conditions prior to a WNV outbreak in the southeast US [46]. Additionally, in the state of New York, wet spring conditions followed by dry summer conditions were associated with increased prevalence of WNV in *Culex* mosquitoes during summer and fall [47].

The results of studies that applied a one-week period to calculate averages of drought indices, or to identify anomalies in drought indices, were less consistent. Tran et al. [43] calculated average anomalies over 8-day periods of modified normalized difference water index (MNDWI; a remote-sensing index that indicates surface moisture and presence of water bodies), relative to long-term averages and standard deviations observed from 2002 to 2011 and found a positive association between MNDWI anomalies in early June, representing above average availability of surface water, and the risk of West Nile fever outbreaks. However, Stilianakis et al. [42] calculated anomalies in weekly mean relative humidity and anomalies in weekly maximum soil water content, relative to weekly data from 1979–2008, and reported significant inverse associations with the odds of human WNV cases for both relative humidity and maximum soil water content during the three weeks leading up to WNV symptom onset. Soverow et al. [40] found that an increase in mean weekly dew point temperature—a measure of humidity—was significantly associated with a 9–38% higher human WNV incidence in the US over the following three weeks. However, the years of potential drought exposure examined by Soverow et al. [40], 2001–2005, are earlier than most exposure years examined by the other included studies, which investigated years up to and including 2018. Notably, the exposure period examined by Soverow et al. [40] does not include the 2011–2012 drought, which was the worst drought recorded in the central US since the 1930s [48]. Furthermore, detection of increased moisture from a heavy rainfall event within the month leading up to a WNV epidemic does not preclude an association between drought and WNV because previous research suggested that human WNV cases are associated with drought conditions 2–6 months prior, and land surface wetting 0.5–1.5 months prior [6].

Two studies included in this review [38,39] were designed to examine if variations in drought conditions over a period of more than one year impacted outbreaks of human WNV (Table 1). Smith et al. [39] found that a warm, dry year preceded by a wet year strongly predicted human WNV cases in the third year in Nebraska. When a drought year precedes a year with high incidence of mosquito-borne illness, the ecological mechanism could be an increased mosquito population due to lower populations of aquatic predators. Because the most efficient aquatic predators of mosquitoes, such as fish, dragonfly nymphs, salamander larvae, and backswimmers [23], have longer developmental periods than their prey, the mosquitoes are often first to re-populate an aquatic habitat that had previously dried out, and under these conditions, vector populations can grow for a period of time without the risk of predation. Skaff and Cheruvelil [38] found that midwestern US counties with a high proportion of semi-permanent wetland that experienced non-drought conditions in the previous year, followed by drought conditions during the focal WNV transmission year had over 150% higher human WNV incidence than similar counties that had experienced any other sequence of annual drought conditions (i.e., non-drought to non-drought, drought to drought, or drought to non-drought). Drought conditions leading to a WNV epidemic in the same year, after a year of normal conditions, may be due to the scarcity of water resources bringing birds and mosquitoes closer together [6], or due to predator avoidance during oviposition by female *Culex* mosquitoes resulting in potentially infected mosquito populations moving closer to residential areas [23].

There are several important covariates to consider when assessing a potential association between drought and human cases of WNV. One of the most important covariates to include is temperature due to its role in reducing the extrinsic incubation period, or the amount of time required before a vector that ingests a WNV-infected bloodmeal can transmit WNV [49]. Most of the nine studies included in this review controlled for the effect of temperature on WNV incidence [7,8,39,40,41,42,43]. Calculations by Smith et al. [39] indicated that while warm temperatures accounted for 29% of WNV cases from 2002–2018 in Nebraska, US, warm temperatures together with drought conditions accounted for 45% of WNV cases. Soverow et al. [40] was the only study with an overall confidence rating of “High” (Figure 3) because it was the only study to control for case-level confounders, such as age and socio-economic status, by using a case-crossover study design. However, the case-crossover approach was designed to assess transient exposures [50], and drought is generally a longer-term exposure. Three of the nine included studies controlled for population immunity to WNV by controlling for cumulative WNV incidence [8,39], or by controlling for the number of WNV outbreaks during the previous year [43]. One study based in Europe and surrounding countries controlled for the effect of being located under bird migration flyways on WNV incidence [43]; this covariate was not assessed by any of the US-based studies.

### Limitations

Our database of existing epidemiological studies on the association between drought and human cases of WNV was limited to nine peer-reviewed articles, and these studies included a wide range of different methodological approaches, drought indices, and study regions. As a result, we decided that it would not be feasible to conduct a meta-analysis on data that was extracted from the included studies.

## 5. Conclusions

The results of studies that examined drought indices averaged over a seasonal period of time [7,8], as well as the results of studies that observed a seasonal pattern of wet spring conditions followed by dry summer conditions prior to a WNV epidemic [37,41], suggest that there could be an ecological mechanism more complex than mosquito dehydration [25] contributing to the association between drought and WNV cases. It is possible that drought increases contact between mosquitoes and birds [6], increasing the WNV infection rate in the vector species, or that predator avoidance during oviposition by female *Culex* mosquitoes brings vector populations into closer contact with humans [23], but more research is needed to confirm these underlying mechanisms across different regions and land use categories. One recent study used ecologically meaningful regions determined by the National Ecological Observatory Network, to examine the relationship between climate data and WNND cases, instead of using state or county boundaries [51]. One limitation of current WNV research is the lack of standardized drought measurement approaches at seasonal timescales. These data would enable better cross-regional comparisons and meta-analyses, which should be prioritized in future studies. Improved understanding of the mechanism by which drought could lead to WNV epidemics would allow for a more standardized approach to assessing this association.

The consistent association between seasonal drought and West Nile virus incidence identified in this review underscores the urgent need for integrated extreme weather-health surveillance systems that can anticipate and respond to emerging zoonotic disease threats. We suggest that incorporating seasonal drought forecasts into their WNV risk assessment frameworks would enable public health departments to be proactive in their implementation of vector control measures and allow for targeted public education campaigns during high-risk periods. Having region-specific clinical guidelines that account for weather-driven variations in WNV transmission intensity will enhance preparedness and would ensure that healthcare providers in drought-prone areas maintain heightened vigilance for neurological symptoms in vulnerable populations, particularly those over 50 years of age.

## Figures and Tables

**Figure 1 microorganisms-13-02851-f001:**
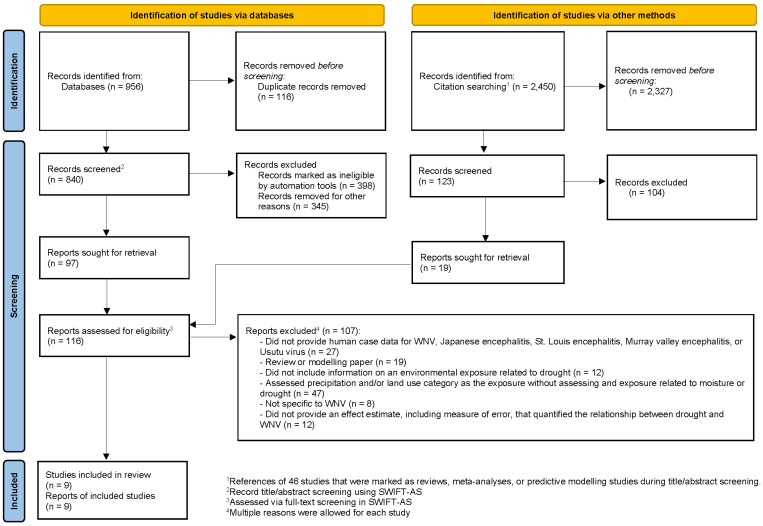
Selection of included studies following PRISMA 2020.

**Figure 2 microorganisms-13-02851-f002:**
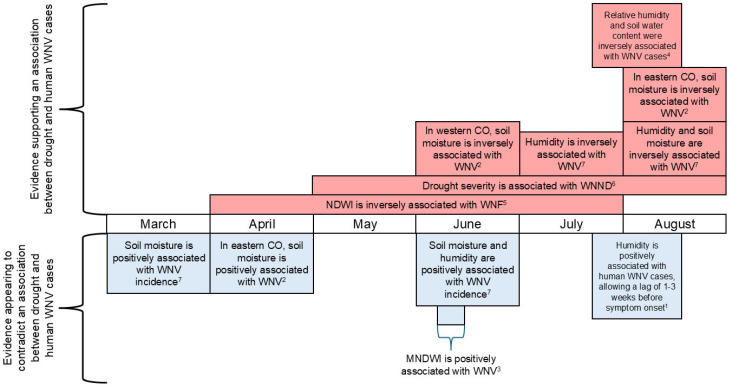
Monthly timeline of exposure variables and key findings for seven studies that investigated whether drought conditions impact human West Nile Virus (WNV) cases during the same year. (NDWI: Normalized Difference Water Index; MNDWI: Modified Normalized Difference Water Index). ^1^ Soverow et al., 2009 [40]; ^2^ Shaman et al., 2010 [37]; ^3^ Tran et al., 2014 [43]; ^4^ Stilianakis et al., 2016 [42]; ^5^ Marcantonio et al., 2015 [7]; ^6^ Paull et al., 2017 [8]; ^7^ Ukawuba & Shaman, 2018 [41].

**Figure 3 microorganisms-13-02851-f003:**
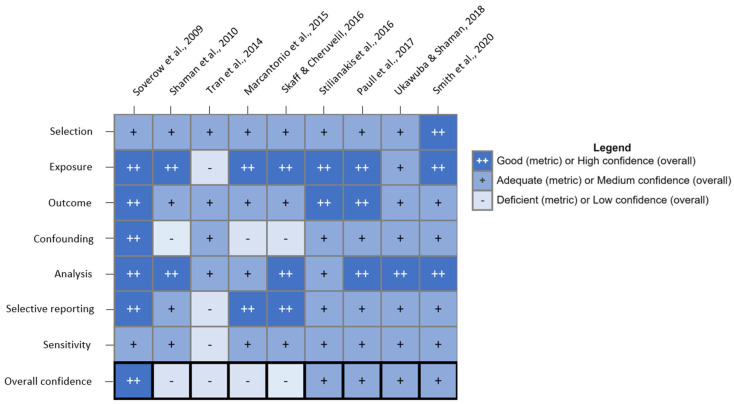
Study Quality Evaluation Results; no study was rated as “Critically deficient” for any of the seven metrics. Soverow et al., 2009 [40]; Shaman et al., 2010 [37]; Tran et al., 2014 [43]; Marcantonio et al., 2015 [7]; Skaff & Cheruvelil, 2016 [38]; Stilianakis et al., 2016 [42]; Paull et al., 2017 [8]; Ukawuba & Shaman, 2018 [41]; Smith et al., 2020 [39].

**Table 1 microorganisms-13-02851-t001:** Summary of key findings from included studies.

Author & Year	Title	Location & Time Period	Lag Between Drought & WNV Cases	Main Findings
Soverow et al., 2009 [40]	Infectious Disease in a Warming World: How Weather Influenced West Nile Virus in the United States (2001–2005)	17 states across the US (county-level data); 2001–2005	Investigates whether drought conditions influence human WNV cases during the same year.	Mean weekly humidity was associated with higher incidence of human WNV cases over the subsequent three weeks.
Shaman et al., 2010 [37]	Hydrologic Conditions Describe West Nile Virus Risk in Colorado	Colorado, US (county-level data); 2002–2007	Investigates whether drought conditions influence human WNV cases during the same year.	In eastern CO, wet spring conditions, followed by dry summer conditions, were associated with human WNV cases.
Tran et al., 2014 [43]	Environmental Predictors of West Nile Fever Risk in Europe	Europe and its neighboring countries; 2002–2013	Investigates whether drought conditions influence human WNV cases during the same year.	Above average surface water during an 8-day period in June was associated with higher numbers of West Nile outbreaks.
Marcantonio et al., 2015 [7]	Identifying the Environmental Conditions Favouring West Nile Virus Outbreaks in Europe	16 countries across western Asia, Europe, and northern Africa; 2010–2012	Investigates whether drought conditions influence human WNV cases during the same year.	Average Normalized Difference Water Index (NDWI) over a 4-month period from April to July was inversely associated with WNF outbreaks.
Skaff and Cheruvelil, 2016 [38]	Fine-scale Wetland Features Mediate Vector and Climate-dependent Macroscale Patterns in Human West Nile Virus Incidence	17 states in the midwestern and northeastern US (county-level data); 2002–2012	Investigates whether interannual variations in drought conditions influence human WNV cases during the second or third year.	*Cx. tarsalis* counties with a high proportional area of semi-permanent wetland that experienced non-drought to drought conditions had a higher incidence of human WNV cases than similar counties that experienced non-drought to non-drought, drought to drought, or drought to non-drought.
Stilianakis et al., 2016 [42]	Identification of Climatic Factors Affecting the Epidemiology of Human West Nile Virus Infections in Northern Greece	Northern Greece; 2010–2014	Investigates whether drought conditions influence human WNV cases during the same year.	Weekly relative humidity and soil water content were both inversely associated with human WNV cases, with a lag time of up to 3 weeks.
Paull et al., 2017 [8]	Drought and Immunity Determine the Intensity of West Nile Virus Epidemics and Climate Change Impacts	Continental US (state-level data); 1999–2013	Investigates whether drought conditions influence human WNND cases during the same year.	Drought measured over a 4-month period (May through August) was the primary climatic driver of increased WNV epidemics, rather than within-season or winter temperatures, or precipitation independently.
Ukawuba and Shaman, 2018 [41]	Association of Spring-summer Hydrology and Meteorology with Human West Nile Virus Infection in West Texas, USA, 2002–2016	West Texas, US (county-level data); 2002–2016	Investigates whether drought conditions influence human WNV cases during the same year	Wet conditions in the spring combined with dry and cool conditions in the summer are associated with increased annual WNV cases.
Smith et al., 2020 [39]	Using Climate to Explain and Predict West Nile Virus Risk in Nebraska	Nebraska, US (county-level data); 2002–2018	Investigates whether variations in monthly drought conditions over a two-year period influence human WNV cases in the third year	A dry year preceded by a wet year was associated with higher incidence of human WNV cases in the following year. Drought accounted for 26% of WNV cases.

## Data Availability

No new data were created or analyzed in this study. Data sharing is not applicable to this article.

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
