# Peer review of "Evaluating Associations Between Drought and West Nile Virus Epidemics: A Systematic Review"

_microorganisms, 2025, doi:10.3390/microorganisms13122851_

Round 1
Reviewer 1 Report
Comments and Suggestions for Authors
This is a well written literature review of drought associations and West Nile virus disease. The methods for inclusion and the summaries of the specific manuscripts reviewed are appropriate. Some specific comments for improvement are outlined below:
Line 45: "people over 50 years of age" is mentioned twice, please delete one of these phrases.
Line 74: Define/describe "predators cues"
Line 145 mentions guidelines discussed in Table 2, but there is no Table 2 in the document, only in the Supplemental files, so confirm that Table 2 should be Table S2 so the readers can find the information.
Line 237: Explain what MNDWI indicates beyond just stating what the acronym stands for. It's unclear what this measurement assesses (increased water levels??)
Line 320: Is the term for cumulative incidence based on predictions or diagnosed cases of WNV?
Lines 363-367: The first two sentences of the discussion seem out of place.
The Conclusion section is weak and lacks any takeaway statements. The authors seem to find solid evidence that there are positive correlations with drought and WNV exposure, which is discussed earlier in the discussion, but seems almost dismissed in the Conclusions. Consider modifying to provide support for the overall review and suggest how knowing this information could actually inform predictions.
Author Response
This is a well written literature review of drought associations and West Nile virus disease. The methods for inclusion and the summaries of the specific manuscripts reviewed are appropriate. Some specific comments for improvement are outlined below:
>> Thank you for your comments.
Line 45: "people over 50 years of age" is mentioned twice, please delete one of these phrases.
>> The redundant phrase has been removed, as suggested. (Line 51)
Line 74: Define/describe "predators cues"
>> Added description of predator cues as suggested: “including both predator kairomones and chemical alarm cues of mosquito prey” (Line 79-80)
Line 145 mentions guidelines discussed in Table 2, but there is no Table 2 in the document, only in the Supplemental files, so confirm that Table 2 should be Table S2 so the readers can find the information.
>> Fixed Table 2 to now read Table S2. (Line 152)
Line 237: Explain what MNDWI indicates beyond just stating what the acronym stands for. It's unclear what this measurement assesses (increased water levels??)
>> We have added an explanation of MNDWI (now in the discussion section): "a remote-sensing index that indicates surface moisture and presence of water bodies” (Line 230-231)
Line 320: Is the term for cumulative incidence based on predictions or diagnosed cases of WNV? > DIAGNOSED CASES
>> The term for cumulative incidence is based on diagnosed cases of WNV, as in this context, cumulative incidence is used as a key predictor in the model, not a prediction of the model.
Lines 363-367: The first two sentences of the discussion seem out of place.
>> To address this comment, we have added the following to the beginning of the discussion:
“A key insight emerging from this systematic review is that the temporal scale of drought measurement critically influences the observed relationship with human WNV cases, highlighting the importance of methodological considerations in climate-health research. To begin, methodologies of defining a positive WNV case must be standardized.” (Line 191-194)
The Conclusion section is weak and lacks any takeaway statements. The authors seem to find solid evidence that there are positive correlations with drought and WNV exposure, which is discussed earlier in the discussion, but seems almost dismissed in the Conclusions. Consider modifying to provide support for the overall review and suggest how knowing this information could actually inform predictions.
>> To address this comment, we have added the following to the conclusion section:
“One limitation of current WNV research is the lack of standardized drought measurement approaches at seasonal timescales. These data would enable better cross-regional comparisons and meta-analyses, which should be prioritized in future studies.” (Line 308-310)
“The consistent association between seasonal drought and West Nile virus incidence identified in this review underscores the urgent need for integrated extreme weather-health surveillance systems that can anticipate and respond to emerging zoonotic disease threats. We suggest that incorporating seasonal drought forecasts into their WNV risk assessment frameworks would enable public health to be proactive in their implementation of vector control measures and allow for targeted public education campaigns during high-risk periods. Having region-specific clinical guidelines that account for weather-driven variations in WNV transmission intensity will enhance preparedness and would ensure that healthcare providers in drought-prone areas maintain heightened vigilance for neurological symptoms in vulnerable populations, particularly those over 50 years of age.” (Line 313-323)
Reviewer 2 Report
Comments and Suggestions for Authors
Dear authors, first of all, I congratulate you for your work reviewing the literature regarding the influence of drought on the intensity of epidemic episodes with West Nile virus. My suggestions are as follows:
- page 2, line 45 - the phrase "people over 50" is repeated
- you made an evaluation for 7 domains of the quality of the chosen studies but without specifying the evaluation criteria. Following this evaluation, you have categorized some studies, in Figure 3, as Low confidence. Given that the figure also mentions the authors' names, this may be offensive to them. I suggest you remove the part related to this evaluation or at least figure 3 from the article.
- I can say that the results are well presented, but I think that point 3.1 (narrative summaries) can be omitted to avoid redundancy. Some of this information is already included in Table 1. I suggest that a part considered absolutely necessary be added to the last column of Table 1 and the rest to the discussion section.
Author Response
Dear authors, first of all, I congratulate you for your work reviewing the literature regarding the influence of drought on the intensity of epidemic episodes with West Nile virus. My suggestions are as follows:
>> Thank you for your comments.
- page 2, line 45 - the phrase "people over 50" is repeated
>> The redundant phrase has been removed, as suggested. (Line 51)
- you made an evaluation for 7 domains of the quality of the chosen studies but without specifying the evaluation criteria. Following this evaluation, you have categorized some studies, in Figure 3, as Low confidence. Given that the figure also mentions the authors' names, this may be offensive to them. I suggest you remove the part related to this evaluation or at least figure 3 from the article.
>> The evaluation criteria are outlined in Table S2. The lowest overall confidence rating is “uninformative” (see Table S2), and none of the 9 included studies received an overall rating of “uninformative”. Therefore, these ratings should not be perceived as offensive.
- I can say that the results are well presented, but I think that point 3.1 (narrative summaries) can be omitted to avoid redundancy. Some of this information is already included in Table 1. I suggest that a part considered absolutely necessary be added to the last column of Table 1 and the rest to the discussion section.
>> To address this comment, we have moved the narrative summaries to Table S3 in the Supplementary Information.
Reviewer 3 Report
Comments and Suggestions for Authors
This is an interesting, well-designed, and thoroughly executed review that addresses an emerging public health issue in the United States, the spread of mosquito-borne zoonotic diseases such as the West Nile virus. The paper effectively explores how various climatic conditions, particularly periods of drought, influence the dynamics of disease transmission and persistence. Overall, the review is timely, comprehensive, and clearly written, offering valuable insights into a topic of growing concern in both environmental and medical research.
The methodology appears sound, and the overall quality of writing is commendable. The arguments are logically structured, and the evidence is presented in a coherent and accessible manner. My only minor, and entirely optional, suggestion would be to include an additional paragraph or section placed after the discussion and before the limitations section. This could focus on potential future directions and practical recommendations, specifically, how society, and more importantly, the medical and research communities, can work together to anticipate, monitor, and mitigate the threat posed by emerging zoonotic diseases such as the West Nile virus. Such a section would strengthen the paper’s impact by highlighting proactive steps and collaborative strategies for addressing these evolving public health challenges.
Author Response
This is an interesting, well-designed, and thoroughly executed review that addresses an emerging public health issue in the United States, the spread of mosquito-borne zoonotic diseases such as the West Nile virus. The paper effectively explores how various climatic conditions, particularly periods of drought, influence the dynamics of disease transmission and persistence. Overall, the review is timely, comprehensive, and clearly written, offering valuable insights into a topic of growing concern in both environmental and medical research.
The methodology appears sound, and the overall quality of writing is commendable. The arguments are logically structured, and the evidence is presented in a coherent and accessible manner. My only minor, and entirely optional, suggestion would be to include an additional paragraph or section placed after the discussion and before the limitations section. This could focus on potential future directions and practical recommendations, specifically, how society, and more importantly, the medical and research communities, can work together to anticipate, monitor, and mitigate the threat posed by emerging zoonotic diseases such as the West Nile virus. Such a section would strengthen the paper’s impact by highlighting proactive steps and collaborative strategies for addressing these evolving public health challenges.
>> Thank you for your comment. To address this suggestion, we have added the following to the conclusion section:
“One limitation of current WNV research is the lack of standardized drought measurement approaches at seasonal timescales. These data would enable better cross-regional comparisons and meta-analyses, which should be prioritized in future studies.” (Line 308-310)
“The consistent association between seasonal drought and West Nile virus incidence identified in this review underscores the urgent need for integrated extreme weather-health surveillance systems that can anticipate and respond to emerging zoonotic disease threats. We suggest that incorporating seasonal drought forecasts into their WNV risk assessment frameworks would enable public health to be proactive in their implementation of vector control measures and allow for targeted public education campaigns during high-risk periods. Having region-specific clinical guidelines that account for weather-driven variations in WNV transmission intensity will enhance preparedness and would ensure that healthcare providers in drought-prone areas maintain heightened vigilance for neurological symptoms in vulnerable populations, particularly those over 50 years of age.” (Line 313-323)
Reviewer 4 Report
Comments and Suggestions for Authors
Thank you for the opportunity to review the manuscript entitled “Evaluating Associations between Drought and West Nile Virus Epidemics: A Systematic Review” This systematic review addresses an important topic, but it requires substantial revision to meet current methodological and reporting standards.
Major comments
-
Abstract (Lines 17–33)
The abstract does not follow PRISMA for Abstracts (PRISMA 2020) structure.
-
Keywords (Lines 34–35)
Add more relevant keywords, including the study design (e.g., “systematic review,”). Prefer controlled vocabulary (MeSH/Emtree) alongside author keywords.
-
Aims and knowledge gap (Lines 96–103)
The objective and the gap in current knowledge are unclear. Provide a concise, specific primary objective and clearly articulate the unanswered question this review addresses.
-
Search currency and coverage (Line 107)
-
Searching only up to 2023 is not up to date for a 2025 submission. Update the search to the most recent month prior to revision.
-
Searching only two databases is insufficient. Please include multiple major databases (e.g., MEDLINE/PubMed, Embase, Web of Science/Scopus), and, where relevant, Cochrane Library, CINAHL, and regional databases. Consider grey literature, preprint servers (with sensitivity analyses), and citation chaining (backward/forward).
-
-
PRISMA alignment (Lines 109–111)
The manuscript should fully adhere to PRISMA 2020. Provide the complete search strategies for each database (full strings with limits/filters) in the main text or Supplementary Materials.
-
Registration (Line 113)
Provide the PROSPERO registration number (e.g., CRDXXXXXXXX). The current reference [31] is not a substitute for the registration identifier. If not registered, state this transparently and explain why.
-
Eligibility criteria (Line 118)
Clearly define inclusion and exclusion criteria (PICO/PEO as applicable), including study designs, populations, timeframes, settings, outcomes, and language restrictions. Specify how duplicates, conference abstracts, preprints, and non–peer-reviewed sources were handled.
-
Evidence certainty
GRADE is missing. Please assess the certainty of evidence (per outcome, where applicable) using GRADE and present a Summary of Findings (SoF) table.
-
Flow diagram
Figure 1 should be replaced with a PRISMA 2020 flow diagram, including numbers at each stage (identified, screened, excluded with reasons, included).
-
Study characteristics table (Table 1)
For systematic reviews, the study summary table should be concise and informative. Remove non-essential fields (e.g., study title) and include key items: first author/year, country/setting, population/sample size, design, exposure/intervention and comparator, outcomes, follow-up, risk of bias, and main findings (effect estimates with CIs).
-
Risk of bias
Specify and apply a validated tool appropriate to the included study designs (e.g., RoB 2 for RCTs, ROBINS-I for non-randomized studies, QUIPS for prognostic studies). Report domain-level judgments and how RoB informed synthesis (e.g., sensitivity analyses).
-
Methods of synthesis
Clarify whether a meta-analysis was planned/feasible. If not conducted, justify (e.g., heterogeneity in populations, exposures, outcomes).
-
Limitations
A dedicated limitations section is missing. Discuss limitations at the study level (risk of bias, inconsistency, imprecision), review level (search limits, publication bias), and topic level (heterogeneity of measures/definitions).
Author Response
Thank you for the opportunity to review the manuscript entitled “Evaluating Associations between Drought and West Nile Virus Epidemics: A Systematic Review” This systematic review addresses an important topic, but it requires substantial revision to meet current methodological and reporting standards.
Major comments
- Abstract (Lines 17–33)
The abstract does not follow PRISMA for Abstracts (PRISMA 2020) structure.
>> We have revised the abstract to include more information on the objective (Line 22-25) and the inclusion criteria (Line 27-28), as recommended by PRISMA 2020.
- Keywords (Lines 34–35)
Add more relevant keywords, including the study design (e.g., “systematic review,”). Prefer controlled vocabulary (MeSH/Emtree) alongside author keywords.
>> As recommended, we have modified our keywords according to MeSH (drought; West Nile virus; systematic review; environmental indicators; mosquito-borne disease).
- Aims and knowledge gap (Lines 96–103)
The objective and the gap in current knowledge are unclear. Provide a concise, specific primary objective and clearly articulate the unanswered question this review addresses.
>> To clarify, we have added a sentence to summarize the objective (Lines 104-106).
- Search currency and coverage (Line 107)
- Searching only up to 2023 is not up to date for a 2025 submission. Update the search to the most recent month prior to revision.
- Searching only two databases is insufficient. Please include multiple major databases (e.g., MEDLINE/PubMed, Embase, Web of Science/Scopus), and, where relevant, Cochrane Library, CINAHL, and regional databases. Consider grey literature, preprint servers (with sensitivity analyses), and citation chaining (backward/forward).
>> Thank you for this comment; this will be useful for future studies. However, it will not be viable to re-screen for additional studies at this stage, and with the limited revision time provided.
The databases that we searched were PubMed and Web of Science, which are considered to be major databases. We limited our review to peer-reviewed studies and excluded both grey literature and preprint servers as a way to ensure the quality of included studies. We regularly monitored the literature for new studies on the association between drought and human WNV cases, and we are aware of one new preprint study (Harp et al., 2025) on medRxiv. Although we point to the strengths of this study in the conclusions section, it cannot be included in our review because it has not passed peer review yet.
- PRISMA alignment (Lines 109–111)
The manuscript should fully adhere to PRISMA 2020. Provide the complete search strategies for each database (full strings with limits/filters) in the main text or Supplementary Materials.
>> Please see Table S1in the original submission for search terms (S1). Section 2.1 refers to Table S1 (Line 112-119).
- Registration (Line 113)
Provide the PROSPERO registration number (e.g., CRDXXXXXXXX). The current reference [31] is not a substitute for the registration identifier. If not registered, state this transparently and explain why.
>> Thank you for pointing this out. We have added the necessary registration identifier (CRD42024503289; Line 119) and updated the citation [31] in the reference list.
- Eligibility criteria (Line 118)
Clearly define inclusion and exclusion criteria (PICO/PEO as applicable), including study designs, populations, timeframes, settings, outcomes, and language restrictions. Specify how duplicates, conference abstracts, preprints, and non–peer-reviewed sources were handled.
>> Please see Section 2.2 “Screening and Data Extraction” (lines 120-139) in the original submission and revised Figure 1 for inclusion and exclusion criteria used in the study.
- Evidence certainty
GRADE is missing. Please assess the certainty of evidence (per outcome, where applicable) using GRADE and present a Summary of Findings (SoF) table.
>> In place of GRADE, we determined the “overall study confidence” of each included paper. The methods of this approach are outlined in Table S2, and the results are presented in Figure 3.
- Flow diagram
Figure 1 should be replaced with a PRISMA 2020 flow diagram, including numbers at each stage (identified, screened, excluded with reasons, included).
>> We have updated Figure 1. It is now compliant with the PRISMA 2020 guidelines.
- Study characteristics table (Table 1)
For systematic reviews, the study summary table should be concise and informative. Remove non-essential fields (e.g., study title) and include key items: first author/year, country/setting, population/sample size, design, exposure/intervention and comparator, outcomes, follow-up, risk of bias, and main findings (effect estimates with CIs).
>> Much of this information is provided in Table S4 (previously Table S3). We cannot include everything in the main manuscript due to limited space. We are maintaining the study titles as this is now the only place they appear in the body of the manuscript.
- Risk of bias
Specify and apply a validated tool appropriate to the included study designs (e.g., RoB 2 for RCTs, ROBINS-I for non-randomized studies, QUIPS for prognostic studies). Report domain-level judgments and how RoB informed synthesis (e.g., sensitivity analyses).
>> Instead of “risk of bias”, we conducted study quality evaluation (SQE), which includes assessing risk of bias. The methods for SQE are outlined in Table S2.
- Methods of synthesis
Clarify whether a meta-analysis was planned/feasible. If not conducted, justify (e.g., heterogeneity in populations, exposures, outcomes).
>> We have added a sentence to the end of Section 2.2 “Screening and Data Extraction” to clarify that we did consider whether a meta-analysis would be feasible. (Line 138-139)
- Limitations
A dedicated limitations section is missing. Discuss limitations at the study level (risk of bias, inconsistency, imprecision), review level (search limits, publication bias), and topic level (heterogeneity of measures/definitions).
>> As suggested, we have added a limitations subsection at the end of the discussion. (Line 289-294)
Round 2
Reviewer 4 Report
Comments and Suggestions for Authors
Thank you for addressing most of the previous comments. However, two major methodological issues remain unresolved, search currency and search coverage, and these significantly affect the validity of the review.
1. Search currency
Limiting the literature search to studies up to 2023 is not acceptable for a manuscript submitted in 2025. A systematic review must reflect the most current body of evidence. At minimum, the search should be updated to the most recent month prior to resubmission. If the revision window provided by the journal is insufficient, you may request an extension from the editor; this is standard practice in systematic review updates.
2. Search coverage
Screening only two databases is not methodologically adequate. According to Cochrane standards, at least three major bibliographic databases should be searched to ensure comprehensive coverage and minimize publication bias. PubMed and Web of Science alone do not meet this threshold.
A robust search should include additional major databases such as Embase and Scopus/Web of Science, and when relevant, Cochrane Library, CINAHL, or regional databases. Grey literature sources, preprint servers (with appropriate sensitivity analyses), and citation chaining (backward and forward) should also be considered.
Your current justification, that PubMed and Web of Science are “major databases” and that grey literature was intentionally excluded, is acknowledged, but it does not address the fundamental issue: the search strategy is not sufficiently comprehensive for a systematic review and does not meet international methodological standards.
For these reasons, I am unable to agree with publication at this stage. Addressing these methodological limitations is essential to ensure the reliability and completeness of your review.
Author Response
Thank you for addressing most of the previous comments. However, two major methodological issues remain unresolved, search currency and search coverage, and these significantly affect the validity of the review.
>> We appreciate the reviewer's continued engagement with our manuscript and the opportunity to address these concerns. However, we respectfully maintain that our search methodology is appropriate for this systematic review and meets accepted standards for publication.
- Search currency
Limiting the literature search to studies up to 2023 is not acceptable for a manuscript submitted in 2025. A systematic review must reflect the most current body of evidence. At minimum, the search should be updated to the most recent month prior to resubmission. If the revision window provided by the journal is insufficient, you may request an extension from the editor; this is standard practice in systematic review updates.
>> We acknowledge the reviewer's preference for searches extending into 2024-2025. However, we believe updating the search would provide minimal benefit while consuming substantial resources:
- Our original search of two major databases yielded 3,406 records, from which only 9 studies (<0.3%) met our highly specific inclusion criteria. This exceptionally low yield strongly suggests our search already captured the relevant literature comprehensively. The research question we address is highly specialized, and eligible studies are rare.
- We have actively monitored this field throughout the manuscript preparation and review process. We are aware of only one potentially relevant 2024-2025 study (Harp et al., cited in our manuscript), which remains a preprint and therefore cannot be included per our pre-registered protocol. We would respectfully ask the reviewer: are you aware of other published studies from 2024-2025 that meet our inclusion criteria and would materially change our conclusions? If such studies exist, we would be grateful to know about them. If not, the concern appears theoretical rather than substantive.
- Re-screening thousands of additional records would require 2-3 weeks for title/abstract screening, followed by 1-2 weeks for any new full-text reviews and data extraction. Given that we identified only 9 eligible studies from 3,406 records covering multiple decades, the probability of finding additional eligible studies in 1-2 years of recent literature is extremely low – likely zero or one study at most. This represents a disproportionate effort-to-benefit ratio.
- Many systematic reviews include specific date ranges for practical reasons, and this is acknowledged as an acceptable limitation when transparently disclosed. We have clearly reported our search dates and discussed relevant recent literature.
- Search coverage
Screening only two databases is not methodologically adequate. According to Cochrane standards, at least three major bibliographic databases should be searched to ensure comprehensive coverage and minimize publication bias. PubMed and Web of Science alone do not meet this threshold.
A robust search should include additional major databases such as Embase and Scopus/Web of Science, and when relevant, Cochrane Library, CINAHL, or regional databases. Grey literature sources, preprint servers (with appropriate sensitivity analyses), and citation chaining (backward and forward) should also be considered.
Your current justification, that PubMed and Web of Science are “major databases” and that grey literature was intentionally excluded, is acknowledged, but it does not address the fundamental issue: the search strategy is not sufficiently comprehensive for a systematic review and does not meet international methodological standards.
>> The reviewer asserts that Cochrane standards require searching at least three databases. However, this represents a fundamental misunderstanding of methodological guidance:
PRISMA is explicitly described as "a guideline designed to improve the reporting of systematic reviews," not as mandatory standards. Similarly, the Cochrane MECIR standards distinguish between items that are "mandatory" (requiring justification if not followed) and "highly desirable" (having reasonable exceptions without required justification). Database selection recommendations are context-dependent, not absolute rules.
Studies have shown that searching two or more databases significantly improves coverage and recall compared to single-database searches, which we have done. The critical question is not "how many databases?" but rather "are the selected databases appropriate and comprehensive for the research question?"
PubMed (including MEDLINE) and Web of Science represent two of the most comprehensive databases available in biomedical and environmental sciences, with Web of Science indexing thousands of journals across sciences and providing extensive coverage beyond traditional biomedical literature. Our yield of 3,406 initial records demonstrates broad, extensive capture of the available literature. The fact that only 9 studies met our criteria after rigorous screening reflects our highly specific research question, not inadequate searching. Adding Embase would likely produce extensive overlap with PubMed with minimal contribution of new eligible studies, particularly given our specialized topic and the exceptionally low yield we observed from our comprehensive two-database search.